# A Difference-of-Convex Functions Approach to Energy-Based Iterative Reasoning

**Daniel Tschernutter**
Infermedica
Graz, Austria
daniel.tschernutter@infermedica.com

**David Diego-Castro**
Infermedica
Gothenburg, Sweden
david.diego-castro@infermedica.com

**Maciej Kasiński**
Infermedica
Wrocław, Poland
maciej.kasinski@infermedica.com

## Abstract

While energy-based models have recently proven to be a powerful framework for learning to reason with neural networks, their practical application is still limited by computational cost. That is, existing methods for energy-based iterative reasoning suffer from computational bottlenecks by relying on expensive optimization routines during training and especially during inference. Furthermore, these routines may not always converge to minimal energy states, potentially leading to suboptimal reasoning. To address these limitations, we propose a novel and efficient algorithm for energy-based iterative reasoning based on a difference-of-convex (DC) functions approach. Our algorithm achieves a significant speedup compared to prior methods while offering theoretical convergence guarantees ensuring locally minimal energy states. In addition, we achieve state-of-the-art or superior performance on continuous reasoning tasks, as demonstrated by our experiments on multiple benchmark datasets from continuous algorithmic reasoning. As such, our method offers a leap in computational efficiency, enabling faster inference with theoretical guarantees, and hence unlocking the potential of energy-based models for iterative reasoning applications.

## 1 Introduction

The human thinking process is described as operating through two distinct modes [31]: the rapid, automatic associations of System 1, and the slower, more controlled symbolic reasoning of System 2. Neural networks have demonstrated remarkable ability to perform System-1-like tasks within well-defined and specific environments. However, when faced with slightly different or harder tasks, neural networks often fail while humans engage in System 2 processes. The latter allows for iterative reasoning about new observations drawing upon prior experience and shared abstractions which remains difficult even for extremely large neural network architectures such as LLMs [35, 51].

There is a variety of recent work that tries to formalize reasoning within a neural network approach, see next section. In this work, we build upon the state-of-the-art in [20, 21] formalizing iterative reasoning as an energy minimization problem, i.e.,

$$\operatorname*{argmin}_{x} E_\theta(x, y) \tag{1}$$

39th Conference on Neural Information Processing Systems (NeurIPS 2025).

for a given problem encoded in $y \in \mathbb{R}^m$ with (partial) solutions $x \in \mathbb{R}^n$. Learning to reason is defined as learning the energy landscape $E_\theta$ parameterized by $\theta$ via

$$\min_\theta \sum_i \left\| \operatorname*{argmin}_x E_\theta(x, y_i) - x_i \right\|^2 \qquad \text{(EMP)}$$

from given problem- and solution-pairs $\{(y_i, x_i) \in \mathbb{R}^m \times \mathbb{R}^n : i \in \{1, \ldots, N\}\}$. Optimization steps from a current (partial) solution $x^k$ to a new $x^{k+1}$ with $E_\theta(x^{k+1}, y) \leq E_\theta(x^k, y)$ are then considered as individual reasoning steps. It has been proven empirically and theoretically that this formulation is superior to direct feed-forward computations, recurrent approaches, and various baselines from neural reasoning in terms of generalization and parameter efficiency [20, 21]. Nevertheless, learning energy landscapes involves solving (1) at training as well as inference time which imposes several limitations on current approaches: (i) Due to the inherent complexity of energy landscapes, heuristics for approximating solutions are used instead of directly solving (1), see next section, which can result in unstable training, (ii) Relying on gradient descent at prediction time is computationally expensive and might hinder practical applications, and (iii) Theoretically, energy-based reasoning yields a natural termination criterion during inference, i.e., an indication to terminate the computation of reasoning steps, by determining if a locally minimal energy state has been found. However, there are no theoretical guarantees that such an energy state is ever reached in previous methods as they rely on gradient descent.

As a remedy, we present a general energy learning framework for continuous iterative reasoning based on difference-of-convex function (DC) optimization. Our main contributions are

1. We introduce a tailored form of energy functions and present a difference-of-convex-function algorithm (DCA), see [4], for (1) powering our novel energy learning algorithm.

2. We derive theoretical convergence guarantees of our DCA routine to local solutions of (1).

3. We show that our DCA routine converges in finitely many steps and, hence, offers a clear termination criterion.

4. Under additional assumptions, we show how our energy learning algorithm can be scaled for batch optimization and present theoretical approximation guarantees for our form of energy function.

## 2 Related Work

**Neural Reasoning.** There is an active area of research that tries to formalize reasoning with neural networks. One group of work builds upon the idea to formulate reasoning as optimization problems and then derives differentiable solvers, e.g., [2, 17, 34, 48, 50]. These approaches are however constrained to tasks of a particular kind, e.g., tasks that can be formulated as quadratic programs [2]. Another group of research formalizes reasoning as iterative computations using neural networks. Following [21], this research can again be broadly subdivided into two areas: one that leverages explicit program representations [26, 36, 38, 10, 13, 37, 55] and another that uses recurrent neural networks [32, 25, 9, 15, 56, 16, 18, 40, 57]. In both areas a challenging problem is to decide when to terminate the computation and has been tackled in various ways usually by learning some sort of halting probabilities [12, 7, 10]. In contrast, our approach naturally imposes a termination criterion by stopping once a local minimum in the energy function has been reached.

**Energy-Based Models.** Energy-based models formulate prediction tasks via energy minimization [33]. That is, external observations $y$ and possible predictions $x$ are both processed by a so-called energy function $E : \mathbb{R}^n \times \mathbb{R}^m \to \mathbb{R}$ which measures how compatible $x$ is with $y$. The convention is that lower energies indicate a higher compatibility. A prediction for $y$ is then defined as a minimum energy state $x^*$ given $y$, see (1). Energy-based models have been used in various ways to learn probabilistic models from data [54, 53, 22, 24, 19, 5, 52]. Our work leverages such energy functions to formalize an iterative reasoning process similar to [20, 21].

**Energy Based Iterative Reasoning.** We are not the first to formulate iterative reasoning as an energy minimization problem. To the best of our knowledge, [20] is the first to formalize reasoning through optimization steps using a general trainable energy function. In particular, the authors make use of a fixed number of gradient steps $T$ with a fixed step size $\lambda$ during training to approximate (1)

within (EMP), i.e., $x_i^T = x_i^{T-1} - \lambda \nabla_x E_\theta(x_i^{T-1}, y_i)$. However, this can lead to unstable training processes as due to potentially complex optimization landscapes it is not guaranteed that $x_i^T$ is a good approximation of a local minimizer of $E_\theta(x, y_i)$. As a remedy, [21] introduced an energy diffusion process in which the authors minimize a sequence of energy landscapes by gradually increasing their complexity and using solutions on previous levels to initialize the gradient descent routine on consecutive levels. The energy landscape is then tuned via a supervised approach on noise corrupted gradients and a contrastive loss component to enforce local minima in the learned energy landscape instead. Nevertheless, both approaches suffer from computational bottlenecks at inference time due to the inherent optimization procedure based on gradient descent and the need for auto-differentiation at test time.

# 3 DC Framework for Energy-Based Reasoning

In this section, we introduce our novel energy-based reasoning framework that builds upon a difference-of-convex functions approach.

## 3.1 DC Energy Landscapes

To allow for sufficient reasoning capabilities an energy function $E_\theta$ should be able to represent a wide range of functional dependencies while at the same time entail structural properties that allow for an efficient solution of $\operatorname{argmin}_x E_\theta(x, y)$ at training time. Based on this observation, we are making use of the following energy function $E_\theta(x, y) = \sum_{i=1}^{N_x} \alpha_i \sigma(\langle w_i, x \rangle + b_i)$, with $\alpha = \alpha_\theta(y)$, $W = W_\theta(y)$, and $b = b_\theta(y)$, where $N_x \in \mathbb{N}$ and $\sigma = \max(\cdot, 0)$ is the ReLU activation function. Note that $E_\theta$ is a single hidden layer neural network in $x$, while its parameters $(\alpha, W, b)$ are again parameterized functions in $y$ with weights $\theta$. In particular, we set $(\alpha_\theta, W_\theta, b_\theta)$ again as single hidden layer neural networks in $y$ with $N_y \in \mathbb{N}$ hidden neurons. Thus, we are using neural networks in $y$ to represent parameters for a neural network in $x$. Due to single hidden layer neural networks being universal approximators [27], we can ensure sufficient representation capabilities for this form of energy function.

As the goal is to learn energy landscapes in a way that minimal-energy states represent solutions of particular reasoning problems, we also want to ensure that such a minimum always exists. Hence, our final definition of $E_\theta$ is as follows

$$E_\theta(x, y) = \frac{\rho}{2} \|x\|^2 - \langle \xi, x \rangle - \omega + \sum_{i=1}^{N_x} \alpha_i \sigma(\langle w_i, x \rangle + b_i), \tag{2}$$

with $\alpha = \alpha_\theta(y)$, $W = W_\theta(y)$, $b = b_\theta(y)$, $\xi = \xi_\theta(y)$, $\rho = \rho_\theta(y) > 0$, and $\omega = \omega_\theta(y)$. Note that we added a general quadratic form, so that for fixed $\theta$ and $y$ we have $E_\theta(x, y) \to \infty$ for $\|x\| \to \infty$. Hence, $E_\theta$ is coercive and continuous in $x$ and thus $\operatorname{argmin}_x E_\theta(x, y) \neq \emptyset$.

Our next result shows that (2) can be decomposed into a difference of convex functions in $x$ for fixed weights $(\rho, \xi, \omega, \alpha, W, b)$, see Lemma 1.

**Lemma 1.** *For fixed weights $(\rho(y), \xi(y), \omega(y), \alpha(y), W(y), b(y))$, the energy function $E_\theta$ is DC in $x$, i.e., $E_\theta(x, y) = E_\theta(x) = g(x) - h(x)$ with*

$$g(x) = \frac{\rho}{2} \|x\|^2 + \sum_{\alpha_i > 0} \alpha_i \sigma(\langle w_i, x \rangle + b_i) \tag{3}$$

$$h(x) = \sum_{\alpha_i < 0} |\alpha_i| \sigma(\langle w_i, x \rangle + b_i) + \langle \xi, x \rangle + \omega, \tag{4}$$

*and $g$, $h$ convex in $x$. For a proof see Appendix A.1* $\qquad\square$

The above DC representation entails desirable properties for our analysis later on. We summarize important characteristics in the following Lemma.

**Lemma 2.** *Let $g$ and $h$ be defined as in Lemma 1, then the following holds*

    *1. $g$ is strongly convex in $x$.*

2. $h$ is (up to a constant) polyhedral convex in $x$, i.e., there exists $q_k \in \mathbb{R}^n$ and $p_k \in \mathbb{R}$ for $k \in \{1, \ldots, K\}$ such that

$$h(x) = \max_{k \in K} \langle q_k, x \rangle + p_k. \tag{5}$$

*For a proof see Appendix A.2* □

## 3.2 Locally Minimal Energy States

This section shows how the above defined energy function can be minimized in $x$ via a tailored DCA. For an introduction to DCA, we refer to [4]. Following [45], the DCA routine for minimizing $E_\theta(x) = g(x) - h(x)$ starting in an arbitrary point $x_0 \in \mathbb{R}^n$ is[1]

$$v \in \partial h(x_k) \tag{6}$$

$$x_{k+1} \in \operatorname*{argmin}_{x} g(x) - \langle v, x \rangle \tag{7}$$

Note that an element in the subgradient of $h$ in (6) is given by

$$\sum_{\alpha_i < 0} |\alpha_i| w_i H(\langle w_i, x_k \rangle + b_i) + \xi \in \partial h(x_k). \tag{8}$$

where $H(z)$ denotes the Heaviside function, i.e., $H(z) = 1$ for $z \geq 0$ and $H(z) = 0$ else. To compute $x_{k+1}$ in the DCA, one has to solve the minimization problem in (7). The following lemma shows that the problem can be equivalently stated as a convex quadratic program.

**Lemma 3.** *The optimization problem*

$$\min_{x \in \mathbb{R}^n} g(x) - \langle v, x \rangle \tag{9}$$

*can be formulated as*

$$\min_{x, z} \frac{1}{2} \begin{pmatrix} x \\ z \end{pmatrix}^T \begin{pmatrix} \rho I & 0 \\ 0 & 0 \end{pmatrix} \begin{pmatrix} x \\ z \end{pmatrix} + \left\langle \begin{pmatrix} x \\ z \end{pmatrix}, \begin{pmatrix} -v \\ \alpha^+ \end{pmatrix} \right\rangle \quad st. \quad \begin{pmatrix} -W^+ & I \\ 0 & I \end{pmatrix} \begin{pmatrix} x \\ z \end{pmatrix} \geq \begin{pmatrix} b^+ \\ 0 \end{pmatrix}, \tag{QP}$$

*where $\alpha^+$ is the vector of strictly positive weights in the output layer, i.e., $\alpha_i^+ = \alpha_{i_j}$ for all $i_j \in \{1, \ldots, N_x\}$ with $\alpha_{i_j} > 0$, $W^+$ is the matrix with rows formed by the corresponding weight vectors $w_{i_j}$, and $b^+$ the vector formed by the corresponding bias terms $b_{i_j}$. For a proof see Appendix A.3.* □

One step in DCA thus simplifies to evaluating Equation (8) and then solving (QP). The next theorem shows that this simple iteration entails favorable convergence properties in the view of supervised learning of minimal energy states.

**Theorem 1.** *Given an arbitrary starting point $x_0$, the DCA routine ((6) and (7)) with $v$ given by Equation (8) and $x_{k+1}$ given as the solution of (QP) converges in finitely many iterations to a DC critical point $x^* \in \mathbb{R}^n$, i.e.,*

$$\partial g(x^*) \cap \partial h(x^*) \neq \emptyset. \tag{10}$$

*Furthermore, $x^*$ is a local minimum of $E_\theta(\cdot, y)$ if and only if*

$$\partial h(x^*) \subseteq \partial g(x^*). \tag{11}$$

*For a proof see Appendix A.4* □

Note that (11) is always fulfilled if $\partial h(x^*)$ or $\partial g(x^*)$ is a singelton. The latter holds true in particular if $\langle w_i, x^* \rangle + b_i \neq 0 \quad \forall i \in \{i : \alpha_i > 0\}$ or $\langle w_i, x^* \rangle + b_i \neq 0 \quad \forall i \in \{i : \alpha_i < 0\}$. If (11) does not hold we can restart the DCA routine with a point $x_0^*$ that yields a strict energy reduction in the first DCA step following [44], see Appendix A.8 for an in-depth discussion on our restart procedure.

---

[1]Here $\partial f$ stands for the subgradient of a convex functions $f$ defined as $\partial f(y) = \{u \in \mathbb{R}^n \mid \forall x \in \mathbb{R}^n : f(x) - f(y) \geq \langle u, x - y \rangle\}$

# 4   Scalable Energy Learning

In theory, the DC framework presented in Section 3 can now be used to learn energy landscapes by supervising the resulting minimal energy states through regression using (EMP) similar to [20]. In particular, one can make use of differentiable convex optimization layers [1] or specialized batched quadratic programming solvers [2] to solve (QP) and run our DCA routine in batches which will converge in finitely many steps due to Theorem 1. Nevertheless, our research shows that (i) relying on differentiable QP solvers, and (ii) the need for our restart routine to ensure local optimality, hinders the ability of our approach to scale to large-scale settings. As a remedy, we introduce additional assumptions on the energy function defined in Equation (2) and show that under these assumptions we can find analytical solutions to (QP) and can guarantee that (11) always holds true for $x^*$, i.e., DCA always converges to a local minimum of the energy function. In addition, in Section 4.1, we show that our energy function approximates a sub class of continuous functions that we call convexly-regular arbitrarily well. We also show that for the univariate prediction case the approximation is universal.

For the remaining part of this work, we make the following additional assumption summarized in Assumption 1.

**Assumption 1.** *Let $\alpha_i \leq 0$ for all $i \in \{1, \ldots, N_x\}$ in Equation (2).*

Note, that this can be easily accomplished by using a non-negative activation function[2] in the neural network component $\alpha_\theta(\cdot)$ and use the resulting values directly in Equation (4). Under this assumption, the following lemma can be derived.

**Lemma 4.** *Let Assumption 1 hold true. Then,* (QP) *can be solved analytically and $x_{k+1} = \frac{1}{\rho} v$. Furthermore, $\partial h(x^*) \subseteq \partial g(x^*)$ always holds true in this case. For a proof see Appendix A.5* $\quad\square$

In the next section we analyze how Assumption 1 affects the approximation capabilities of our energy function, as indeed restricting shallow neural networks to only positive weights can make them loose their universal approximation guarantees [49].

## 4.1   Approximation Guarantees

In our energy function (2) the weights to parameterize the function in $x$, i.e., $(\rho_\theta(y), \xi_\theta(y), \omega_\theta(y), \alpha_\theta(y), W_\theta(y), b_\theta(y))$, are all single hidden layer neural networks in $y$, and hence universal approximators [27], see also Appendix A.6. We thus focus the following analysis on approximations of functions in $x$ and need the following definitions and results from earlier work.

**Definition 1.** *Let $\mathbb{X} \subseteq \mathbb{R}^n$ be convex. A function $f : \mathbb{X} \to \mathbb{R}$ is called $\rho$-weakly-convex, or simply weakly-convex, if there exists a $\rho > 0$ and a convex function $h$ such that $f + \rho/2\|\cdot\|^2 = h$. The function is called weakly-concave if $-f$ is weakly-convex. The set of all weakly-convex functions in $\mathbb{X}$ is denoted by $\mathcal{WC}(\mathbb{X})$.*

It can be shown that weakly-convex functions are universal approximators. To formalize this claim recall that $\mathcal{C}_0(\mathbb{X})$ denotes the set of continuous functions that vanish at infinity, i.e., for all $\epsilon > 0$ there exists a compact set $K \subseteq \mathbb{X}$ such that $|f(x)| < \epsilon$ if $x \notin K$. Then, the following holds.

**Lemma 5** (Theorem 6 in [43]). *Let $\mathbb{X} \subseteq \mathbb{R}^n$ be closed (or open) convex. Then, $\mathcal{WC}(\mathbb{X}) \cap \mathcal{C}_0(\mathbb{X})$ is dense in $\mathcal{C}_0(\mathbb{X})$ equipped with the infinity norm. This statement also holds for weakly-concave functions (by switching signs).*

Furthermore, weakly-convex functions can be represented in a special form, see Lemma 6.

**Lemma 6** (Theorem 3 in [43]). *A (closed) function $f$ is $\rho$-weakly-convex if and only if $f(x) = \sup_{t \in T} \langle q_t, x \rangle + p_t - \rho/2\|x\|^2$ for some (not necessarily finite) index set $T$.*[3]

Now, by combining Lemma 2 and Lemma 6 we see that our energy function is weakly-concave in $x$, i.e., it has exactly the form

$$\frac{\rho}{2}\|x\|^2 - \max_k \langle q_k, x \rangle + p_k. \tag{12}$$

---

[2] In our implementation, we are using softplus activations for both $\alpha_\theta(\cdot)$ and $\rho_\theta(\cdot)$.

[3] Note that Theorem 3 in [43] is stating that $f(x) = \sup_{t \in T} \langle a_t, x \rangle + b_t + \sigma\|x\|^2$ as the authors use $\sigma = -\rho$ to denote the modulus of convexity.

Note, however, that it is not immediately clear that also any weakly-concave function can be approximated by our energy function as the vectors $q_k$ and biases $p_k$ in Lemma 2 follow a special form. Nevertheless, if this is the case or equivalently if we are able to prove that $h$ can approximate continuous convex functions, Lemma 5 yields theoretical approximation guarantees for our energy function.

To derive sufficient conditions for $h$ to be able to approximate a continuous convex function, we note that we merely impose sign constraints on the output layer of the involved shallow neural network. Thus, $h$ can be seen as a single layer input convex neural network (ICNN) with weighted input skip connections in $x$ [23]. Input skip connections were introduced for ICNNs in [3] to increase their expressivity by allowing identity mappings between layers as otherwise the non-negativity constraint would be too restrictive. Indeed, it has then been shown that ICNNs are able to approximate arbitrary continuous convex functions on compact convex domains (see, e.g., Theorem 1 in [14] or Proposition 3 in [28]). However, those approximation guarantees require deeper ICNNs while $h$ is merely a single layer ICNN. As pointed out by [23], the derivations in [14, 28] are merely for theoretical purposes as they require as many layers as affine pieces of the piecewise linear convex function they are trying to approximate and only a single neuron per layer. The authors, thus, derive the following result.

**Lemma 7** (Corollary 4.8 and Proposition 4.9 in [23]). *A convex function implemented by a single hidden layer ReLU network (with or without weighted input skip connections) can also be implemented by a single hidden layer ICNN with the same width.*

The question if $h$ can approximate a continuous convex function thus boils down to the question if it can be approximated by a convex shallow ReLU network. Hence, we define the following

**Definition 2.** *Let $\mathbb{X} \subseteq \mathbb{R}^n$ be convex and compact. We call a continuous convex function $r : \mathbb{X} \to \mathbb{R}$ $\epsilon$-convexly-ReLU-representable if there exists a convex single layer ReLU network (with or without weighted input skip connections) $\mathcal{NN}$ such that $\|\mathcal{NN} - r\|_\infty < \epsilon$.*

Given a function $f \in \mathcal{C}_0(\mathbb{X})$, we know from Lemma 5 that for all $\epsilon > 0$ there exists a $\rho$-weakly-concave function $f_\epsilon$ with $\|f - f_\epsilon\|_\infty < \epsilon$. Then we know that $r = -f_\epsilon + \rho/2\|x\|^2$ is convex and we make the following definition.

**Definition 3.** *Let $\mathbb{X} \subseteq \mathbb{R}^n$ be convex and compact and let $f \in \mathcal{C}_0(\mathbb{X})$. We call $f$ convexly-regular, if we can always choose $f_\epsilon$ such that $r$ is $\epsilon$-convexly-ReLU-representable.*

Now, Theorem 2 summarizes the above derivations

**Theorem 2.** *Let $\mathbb{X} \subseteq \mathbb{R}^n$ be convex and compact. Under Assumption 1 every convexly-regular function $f \in \mathcal{C}_0(\mathbb{X})$ can be approximated arbitrarily well by $E_\theta$ as a function in $x$.*

*Proof.* Let $\epsilon > 0$ be arbitrary and $\hat{\epsilon} = \epsilon/2$. From Lemma 5 we know that there exists a weakly-concave function $f_{\hat{\epsilon}}$ such that $\|f - f_{\hat{\epsilon}}\|_\infty < \hat{\epsilon}$. Hence, there exists a $\rho > 0$ and a convex function $r$ such that $-f_{\hat{\epsilon}}(x) + \rho/2\|x\|^2 = r(x)$. Furthermore, Definition 3 ensures that we can always choose $f_{\hat{\epsilon}}$ such that $r$ is $\hat{\epsilon}$-convexly-ReLU-representable. Thus, there exists a convex ReLU neural network $\mathcal{NN}$ with $\|r - \mathcal{NN}\|_\infty < \hat{\epsilon}$. From Lemma 7, we know that there exist $\alpha \leq 0$, $W$, $b$, $\xi$, and $\omega$ such that $\mathcal{NN}(x) = \sum_i |\alpha_i| \sigma(\langle w_i, x \rangle + b_i) + \langle \xi, x \rangle + \omega$ which we define as $h$. Hence, for $E_\theta(x) = \frac{\rho}{2}\|x\|^2 - h(x)$ we have

$$\|f - E_\theta(x)\| = \left\| f - \left( \frac{\rho}{2}\|x\|^2 - h(x) \right) \right\|_\infty \leq \|f - f_{\hat{\epsilon}}\|_\infty + \left\| f_{\hat{\epsilon}} - \left( \frac{\rho}{2}\|x\|^2 - h(x) \right) \right\|_\infty \quad (13)$$

$$< \hat{\epsilon} + \|h - r\|_\infty < 2\hat{\epsilon} = \epsilon. \quad (14)$$

$\square$

Definition 3 is rather technical and to the best of our knowledge there are no general results on conditions under which a convex function can be approximated by a single layer ReLU network which is itself convex. However, the following theorem shows that the class of convexly-regular functions is sufficiently large in the univariate case.

**Theorem 3.** *Let $\mathbb{X} \subseteq \mathbb{R}$ be compact and convex. Then, every $f \in \mathcal{C}_0(\mathbb{X})$ is convexly-regular. For a proof see Appendix A.7* $\square$

## 4.2 Pseudocode

We now combine our derivations in Section 3 and Assumption 1 to define our algorithm for scalable energy learning via a batched DCA approach named **DCAReasoner**.[4] A pseudocode is presented in Algorithm 1.

---

**Algorithm 1:** DCAReasoner: Scalable Energy Learning via Batched DCA

---
**Data:** $(y_i, x_i) \in \mathbb{R}^m \times \mathbb{R}^n$, lower and upper bounds for starting points $l, u \in \mathbb{R}^n$
**Result:** $E_\theta(\cdot, \cdot)$

1 **while** *not converged* **do**
2      Sample batch of data $(y_j, x_j)_{j \in B}$;
3      Perform forward pass for parameters $(\alpha, W, b, \rho, \xi) \leftarrow (\alpha, W, b, \rho, \xi)((y_j)_{j \in B})$;
4      Sample uniformly random starting points $(x_j^0)_{j \in B} \sim \mathcal{U}(l, u)$;
5      $k \leftarrow 0$;
     // Initializing $x_j^{k+1}$ in a meaningful way
6      **while** $\max_{j \in B} \|x_j^k - x_j^{k+1}\| > tol$ **do**
7          $x^{k+1} \leftarrow \frac{1}{\rho}\left(\sum_i |\alpha_i| w_i H(\langle w_i, x^k \rangle + b_i) + \xi\right)$;
8          $k \leftarrow k + 1$;
9      **end**
10      Update $\theta$ using Adam and $\nabla_\theta \sum_{j \in B} \|x_j^k - x_j\|^2 / |B|$;
11 **end**

---

Note that we used $\max_{j \in B} \|x_j^k - x_j^{k+1}\| > tol$ as a stopping criterion for the DCA routine. Empirically, we observe that a few DCA iterations ($< 10$) are enough for the whole batch to converge for $tol = 10^{-5}$. Indeed, most of the time we observe $\max_{j \in B} \|x_j^k - x_j^{k+1}\| \ll tol$ approaching machine precision, and hence, the finite convergence property can also be observed empirically. See Figure 2 in Appendix B.1 for an illustration of how the norm differences decrease to zero with DCA iterations for a batch size of $512$, i.e., when solving $512$ reasoning tasks in parallel. We also note that we skipped the neural network for the bias term, i.e., $\omega(y)$, in Algorithm 1 as it does not change the local minimizer of (1) and was merely used for our theoretical derivations. Furthermore, the network for $b$ will not be updated during training as a consequence of the Heaviside function. Nevertheless, similar partial optimization routines, in which parts of the parameters are randomly initialized and then frozen, have been successfully applied in neural learning, see e.g. [29].

## 5 Numerical Experiments

### 5.1 Experimental Setup

We first evaluate our algorithm on five continuous algorithmic reasoning benchmark datasets from earlier research [20, 21] in Section 5.2. All tasks are aiming to capture different aspects of reasoning. We report the mean squared errors, as well as, inference times of our DCAReasoner and two state-of-the-art baselines from energy-based iterative reasoning. The evaluation is performed on 10000 test problems and repeated five times to report the mean and standard error of our metrics. Having established that our algorithm is superior (or on par) with state-of-the-art but significantly faster, we then demonstrate how our DCAReasoner might unlock reasoning capabilities in language models by learning energy landscapes in token embedding spaces in Section 5.3.

### 5.2 Continuous Algorithmic Reasoning Benchmarks

**Baselines.** We consider two baselines from state-of-the-art energy based iterative reasoning: (i) *Energy based reasoning through energy minimization (IREM)*: This baseline learns an energy function minimizing (EMP) by approximating $\operatorname{argmin}_x E_\theta(x, y)$ via a fixed number of gradient steps [20]. During inference it uses again a subgradient descent method but leveraging a greater number of steps than during training.(ii) *Energy based reasoning through energy diffusion (IRED)*: Here, the idea is to minimize a sequence of energy landscapes gradually increasing their complexity and using solutions on previous levels to initialize the gradient descent routine on consecutive levels [21]. The energy

---

[4]`https://github.com/DanielTschernutter/DCAReasoner`

| Dataset | DCAReasoner (Ours) | | IRED | | IREM | |
|---|---|---|---|---|---|---|
| | MSE | Inference-Time [s] | MSE | Inference-Time [s] | MSE | Inference-Time [s] |
| *Same Difficulty* | | | | | | |
| Matrix Inverse | **0.0096±0.0000** | 2.6189±0.0305 | 0.0097±0.0000 | 33.7056±0.8818 | 0.0101±0.0000 | 22.6199±0.6127 |
| Matrix Completion | **0.0177±0.0000** | 1.3373±0.0125 | 0.0179±0.0000 | 33.6597±0.9125 | 0.0180±0.0000 | 22.7441±0.5296 |
| Parity | **0.0301±0.0003** | 0.6053±0.0381 | 0.4859±0.0026 | 9.1011±0.2809 | 0.2504±0.0001 | 1.9797±0.1463 |
| QR Decomposition | **0.1438±0.0001** | 2.3051±0.0261 | 0.2175±0.0001 | 48.1915±1.4199 | 0.1521±0.0001 | 36.2775±1.0231 |
| Matrix Multiplication | **0.0480±0.0000** | 1.6790±0.0258 | 0.0919±0.0000 | 34.1602±0.8609 | 0.0903±0.0000 | 23.6510±0.6229 |
| *Harder Difficulty* | | | | | | |
| Matrix Inverse | 0.2077±0.0003 | 2.6017±0.0226 | 0.2064±0.0003 | 33.3662±0.6442 | **0.2063±0.0006** | 22.7757±0.4990 |
| Matrix Completion | 0.2100±0.0001 | 1.3491±0.0211 | 0.2094±0.0002 | 33.2705±0.7230 | **0.2058±0.0002** | 22.9266±0.5348 |
| Parity | **0.0301±0.0003** | 0.5863±0.0067 | 0.4885±0.0010 | 8.6239±0.0879 | 0.2504±0.0001 | 1.9477±0.0983 |
| QR Decomposition | **0.8847±0.0003** | 2.3374±0.1298 | 1.0376±0.0002 | 47.7211±1.0362 | 1.3267±0.0006 | 35.3764±0.8735 |
| Matrix Multiplication | **0.2974±0.0003** | 1.6877±0.0243 | 0.4506±0.0002 | 33.3451±0.6479 | 0.4524±0.0002 | 23.5748±0.6184 |

Table 1: Evaluation on continuous algorithmic reasoning tasks. Models are evaluated on test problems drawn from the training distribution (same difficulty) and a harder test distribution (harder difficulty). We report the mean squared error and the inference time. We perform five evaluation runs and report the mean and standard errors.

landscape is then tuned by supervising on noise corrupted gradients and a contrastive loss component to enforce local minima in the energy landscape. We scale the network size in our baselines to ensure that each of them has roughly the same number of parameters. For a detailed discussion of our experimental setup see Appendix B.2.

**Datasets.** In our experiments we evaluate all baselines based on five datasets from earlier research [20, 21]. Each of them is evaluated once with the same level of difficulty, i.e., test cases are drawn from the training distribution, and once with a harder level of difficulty, in which test cases are drawn from a problem specific harder test distribution following [20, 21]. The latter should test the algorithms ability to generalize reasoning capabilities to new unseen problem settings. In particular, we use: (i) Matrix Inverse: The task is to invert a random $20 \times 20$ matrix. It aims at testing numerical reasoning. Harder problems are created by creating less well-conditioned matrices to invert. (ii) Matrix Completion: The task is to recover masked out values in a random low-ranked $20 \times 20$ matrix constructed from two low-rank matrices $U$ and $V$. Harder tasks are created by increasing the complexity of $U$ and $V$. It aims at both structural and analogical reasoning. (iii) Parity: Given a random vector in $[0, 1]^{20}$ the task is to decide whether or not the number of entries which are greater than 0.5 is odd or even, i.e., the target is 0 for even 1 for odd, see also [25]. Harder tasks are created by increasing the magnitude of vector entries. (iv) QR Decomposition: The task is to compute the QR decomposition of a random $20 \times 20$ matrix with entries in $[-1, 1]$. Harder problems are created by changing the magnitude of matrix entries. (v) Matrix Multiplication: Given a random $20 \times 20$ matrix $M$ the task is to compute the square, i.e., $M^2$. Harder problems are created by changing the magnitude of matrix entries. For an in-depth discussion of our benchmark datasets see Appendix B.3.

**Results.** Our results are summarized in Table 1. In terms of mean squared error, DCAReasoner is mostly on par with IRED and IREM on the matrix inverse and the matrix completion dataset, while we see larger improvements for the remaining datasets. Noteworthy, we observe a decrease in MSE by a factor of $\sim 10$ on the parity dataset, which might stem from the fact that the prediction is univariate in this case, for which we have established universal approximation guarantees in Theorems 2 and 3. In terms of inference time, we see large improvements of factors between 14 and 27 for IRED and between 3 and 18 for IREM. Furthermore, we performed additional experiments showing that our predictions are robust to noisy input data on the example of the QR Decomposition dataset. All details are reported in Appendix B.4.

### 5.3 Energy-Based Reasoning in Token Embedding Space

In the last section we have empirically proven that our algorithm yields state-of-the-art performance results but is significantly faster at inference time than previous energy-based models for iterative reasoning. Furthermore, it offers theoretical convergence guarantees and performs well in high-dimensional settings. As such, we think that our algorithm might be used to improve reasoning skills of language models by learning energy landscapes in token embedding spaces which might be used during inference for energy-guided text generation. Note that our baselines are not well-suited for such a setting as large inference times especially in high-dimensional token embedding spaces might considerably slow down token generation in practice.

| Alg. | MSE | Accuracy[%] | Inference Time[%] |
|------|-----|-------------|-------------------|
| IREM | 0.012 | **96%** | 336% |
| IRED | 0.027 | **96%** | 1330% |
| DCAReasoner | **0.008** | **96%** | **100%** |

Table 2: Test evaluation performance on text classification task. We report the mean squared error of the prediction and the embedding of the target text. Accuracy is computed by using the target closest to the prediction. Inference time is reported in percent, with 100% indicating the lowest time.

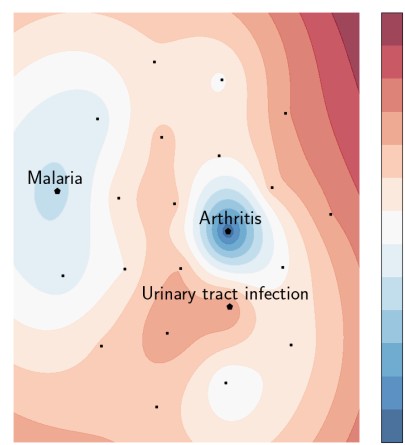

Figure 1: Visualization of Energy Landscape in Token Embedding Space for the Input Sentence "*My muscles are weak, my neck is stiff, and my joints are swollen. I can't move around very well, and walking is really painful.*".

As a fully developed approach for energy-guided text generation to improve reasoning is out of the scope of this work, we demonstrate how DCAReasoner is able to learn reasonable energy-landscapes in token embedding spaces in a simpler setting. In particular, we make use of the symptom-to-diagnosis dataset for medical reasoning which is freely available on Hugging Face.[5] It provides a training and test dataset consisting of short texts in which a patient describes her symptoms and a corresponding diagnosis out of a set of 22 medical diagnoses. We then use the CLS token embeddings of a finetuned uncased DistilBERT model of those texts as inputs $y$ and the embeddings of the corresponding diagnosis as a ground truth $x$ to train the DCAReasoner and our baselines in a continuous reasoning setting. More details of our experiments are reported in Appendix B.5.

We summarize the results in Table 2. Our algorithm yields the lowest mean squared error more than three times faster than IREM and thirteen times faster than IRED. We also visualize the energy landscape in token embedding space[6] learned by our algorithm in Figure 1 using as an example the sentence "*My muscles are weak, my neck is stiff, and my joints are swollen. I can't move around very well, and walking is really painful.*" with diagnosis *arthritis* from the test set. Note that *arthritis* has indeed the lowest energy, while embedding vectors of diseases like *malaria* which shares symptoms like weakness and joint pain are also assigned lower energy values. Furthermore, seemingly unrelated diagnoses like *urinary tract infection* yield higher energy values, indicating that our algorithm is capable of learning reasonable energy landscapes in the token embedding space.

## 6 Conclusion

We proposed a new algorithm named DCAReasoner for energy-based continuous iterative reasoning. It is build upon a tailored class of energy functions for which we derived theoretical approximation guarantees. In addition, we presented theoretical convergence guarantees for the inherent DCA routine. That is, we showed that it converges to local minima in finitely many steps independent of

---

[5]`https://huggingface.co/datasets/gretelai/symptom_to_diagnosis`, the dataset is licensed under Apache 2.0

[6]We used t-SNE to visualize the 768 dimensional embedding vectors in the plane. Furthermore, we computed the energies for all 22 target embeddings and interpolated the results using a radial basis function interpolator.

the starting point. Empirically, we have proven that it yields improved performance and inference times.

**Limitations.** Our DCAReasoner shows promising results for neural iterative reasoning. However, there are still limitations. First, our DCAReasoner, as presented in this work, does not yet make use of any external memory. That is, DCAReasoner cannot store intermediate results during reasoning which might be beneficial in some reasoning tasks. Second, the form of our energy function might require a large number of trainable parameters if a large number of hidden neurons in $x$, i.e., $N_x$, is desirable. For instance, the mapping $W_\theta(y)$ requires $m \cdot N_y + N_y \cdot N_x \cdot n$ trainable parameters. Empirically, we see however that smaller values for $N_x$ and larger values for $N_y$ are sufficient (our experiments use $N_x = 8$ and $N_y = 4000$). Third, in the current form DCAReasoner is designed for continuous reasoning tasks and hence cannot directly handle discrete reasoning tasks. However, we note that earlier work on energy-based reasoning shows how discrete tasks, formulated in a continuous setting, yield promising results [20, 21]. Furthermore, we conducted preliminary experiments on a discrete task, specifically solving Sudoku puzzles using the dataset provided in [41], and report our early results in Appendix C.

**Societal Impacts.** To the best of our knowledge, there are no immediate positive or negative social impacts that can be derived from our work in the current form.

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

# A   Mathematical Appendix

## A.1   Proof of Lemma 1

It is clear that $E_\theta(x, y)$ can be decomposed as

$$E_\theta(x, y) = E_\theta(x) = g(x) - h(x) \tag{15}$$

with

$$g(x) = \frac{\rho}{2}\|x\|^2 + \sum_{\alpha_i > 0} \alpha_i \sigma(\langle w_i, x \rangle + b_i) \tag{16}$$

$$h(x) = \sum_{\alpha_i < 0} |\alpha_i| \sigma(\langle w_i, x \rangle + b_i) + \langle \xi, x \rangle + \omega. \tag{17}$$

Furthermore, $\langle w_i, x \rangle + b_i$ is affine in $x$ and hence convex and $\sigma$ is non-decreasing and convex. Thus the composition $\sigma(\langle w_i, x \rangle + b_i)$ is always convex and the claim follows by observing that $g$ and $h$ are linear combinations of convex functions with positive weights.

## A.2   Proof of Lemma 2

For point one, note that the modulus of strong convexity is defined as

$$\rho(g) = \sup_{\rho \geq 0}\{g(x) - \frac{\rho}{2}\|x\|^2 \text{ is convex}\}, \tag{18}$$

see, e.g., [44]. It is easy to see that $g - \frac{\rho}{2}\|\cdot\|^2$ is convex and hence $\rho(g) \geq \rho > 0$, which implies strong convexity.

For point two, polyhedral convex functions can be characterized as

$$\max\{\langle q_k, x \rangle + p_k : k \in \{1, \dots, K\}\}, \tag{19}$$

see, e.g., [39]. Note that

$$h(x) - \omega = \sum_{\alpha_i < 0} |\alpha_i| \sigma(\langle w_i, x \rangle + b_i) + \langle \xi, x \rangle \tag{20}$$

$$= \sum_{\alpha_i < 0} \sigma(\langle |\alpha_i| w_i, x \rangle + |\alpha_i| b_i) + \langle \xi, x \rangle \tag{21}$$

$$= \max_{\delta_i \in \{0,1\}} \sum_{\alpha_i < 0} \delta_i (\langle |\alpha_i| w_i, x \rangle + |\alpha_i| b_i) + \langle \xi, x \rangle, \tag{22}$$

where the last equality follows from the fact that the maximum is achieved in the case where $\delta_i = 1$ if $\langle |\alpha_i| w_i, x \rangle + |\alpha_i| b_i \geq 0$ and $\delta_i = 0$ else. Furthermore,

$$\max_{\delta_i \in \{0,1\}} \sum_{\alpha_i < 0} \delta_i (\langle |\alpha_i| w_i, x \rangle + |\alpha_i| b_i) + \langle \xi, x \rangle = \tag{23}$$

$$\max_{\delta_i \in \{0,1\}} \left\langle \sum_{\alpha_i < 0} |\alpha_i| \delta_i w_i + \xi, x \right\rangle + \sum_{\alpha_i < 0} |\alpha_i| \delta_i b_i, \tag{24}$$

which proves the claim.

## A.3   Proof of Lemma 3

First, note that

$$g(x) - \langle v, x \rangle = \frac{\rho}{2}\|x\|^2 + \sum_{\alpha_i > 0} \alpha_i \sigma(\langle w_i, x \rangle + b_i) - \langle v, x \rangle, \tag{25}$$

which can be equivalently stated as

$$\min_{x,z} \frac{1}{2} \begin{pmatrix} x \\ z \end{pmatrix}^T \begin{pmatrix} \rho I & 0 \\ 0 & 0 \end{pmatrix} \begin{pmatrix} x \\ z \end{pmatrix} + \left\langle \begin{pmatrix} x \\ z \end{pmatrix}, \begin{pmatrix} -v \\ \alpha^+ \end{pmatrix} \right\rangle, \tag{26}$$

and the non-linear equality constrained $z_i = \sigma(\langle w_i, x \rangle + b_i)$ for all $i \in \{1, \dots, N_x\}$ with $\alpha_i > 0$. Now, the inequality constraints

$$z_i \geq 0 \tag{27}$$
$$z_i \geq \langle w_i, x \rangle + b_i \tag{28}$$

imply that $z_i \geq \sigma(\langle w_i, x \rangle + b_i)$. As the only term in the objective function containing $z$ is given by $\langle \alpha^+, z \rangle$ and $\alpha^+$ is elementwise strictly positive, we have that problem (26) with constraints (27) and (28) always results in $z_i^* = \sigma(\langle w_i, x^* \rangle + b_i)$ at an optimal point $(x^*, z^*)$. Hence, the claim follows.

### A.4 Proof of Theorem 1

From Lemma 2, we know that $g$ is strongly convex with $\rho(g) \geq \rho$. Convergence thus follows by Theorem 6 in [45]. Furthermore, Lemma 2 shows that $h$ is polyhedral convex, which proves the finite convergence property using Theorem 9 in [45]. Last, the fact that $x^*$ is a local minimum if Equation (11) holds true follows from Corollary 2 in [45].

### A.5 Proof of Lemma 4

It is easy to see that under Assumption 1, the function $g$ reduces to $\frac{\rho}{2}\|x\|^2$. Hence (QP) reduces to

$$\frac{\rho}{2}\|x\|^2 - \langle x, v \rangle, \tag{29}$$

with the global solution $\frac{1}{\rho} v$. Furthermore, in this case $g$ is differentiable and hence $\partial g(x^*) = \{\nabla g(x^*)\}$ is a singleton. Thus, Equation (11) always holds true under Assumption 1.

### A.6 Details on Approximation in $y$

In our energy function (2) the weights to parameterize the function in $x$, i.e., $(\rho_\theta(y), \xi_\theta(y), \omega_\theta(y), \alpha_\theta(y), W_\theta(y), b_\theta(y))$, are all single hidden layer neural networks in $y$. Since $(\xi_\theta(y), \omega_\theta(y), W_\theta(y), b_\theta(y))$ are all using the identity as an output layer activation those are universal approximators [27]. Note also that $\rho_\theta(y)$ and $\alpha_\theta(y)$ are using softplus activations in the output layer. Hence, they are still able to approximate any strictly positive continuous function arbitrarily well as shown in the following.

Let $f$ be an arbitrary strictly positive and continuous function. Then $\log(e^f - 1)$ is a continuous function mapping to $(-\infty, \infty)$. As such it can be approximated arbitrarily well by a single hidden layer network by [27]. Further, if we now equip this network with a softplus activation, we approximate softplus $\left(\log(e^f - 1)\right) = f$ by using that softplus is Lipschitz. We formalize this in the following lemma.

**Lemma 8.** *Let $f : \mathbb{R}^n \to (0, \infty)$ be continuous. Then for every $\epsilon > 0$ there is a single hidden layer network with a softplus activation, $\nu_\epsilon$, such that $\|f - \nu_\epsilon\|_\infty < \epsilon$*

*Proof.* Let $\epsilon > 0$. By [27], there is a single hidden layer network $N_\epsilon$ such that $\|\log\left(e^{f(\cdot)} - 1\right) - N_\epsilon\|_\infty < \epsilon$. Using that the function $\text{softplus}(y) = \log(1 + e^y)$ is Lipschitz of constant 1 and that $z \mapsto \log(e^z - 1)$ is its inverse, we deduce that for every $x \in \mathbb{R}^n$: $\left|f(x) - \log\left(1 + e^{N_\epsilon(x)}\right)\right| \leq \left|\log\left(e^{f(x)} - 1\right) - N_\epsilon(x)\right|$ and therefore

$$\|f - \log\left(1 + e^{N_\epsilon}\right)\|_\infty = \sup_{x \in \mathbb{R}^n} \left|f(x) - \log\left(1 + e^{N_\epsilon(x)}\right)\right|$$

$$\leq \sup_{x \in \mathbb{R}^n} \left|\log\left(e^{f(x)} - 1\right) - N_\epsilon(x)\right| = \|\log\left(e^f - 1\right) - N_\epsilon\|_\infty < \epsilon$$

By setting $\nu_\epsilon(x) = \log\left(1 + e^{N_\epsilon(x)}\right)$, the claim follows. □

### A.7 Proof of Theorem 3

Let $f \in \mathcal{C}_0(\mathbb{X})$ and $\epsilon > 0$. From Lemma 5 we know that there exists a weakly-concave function $f_\epsilon$ such that $\|f - f_\epsilon\|_\infty < \epsilon$. Hence, there exists a $\rho > 0$ and a convex function $r$ such that $-f_\epsilon(x) + \rho/2\|x\|^2 = r(x)$. We need to show that $r$ is $\epsilon$-convexly-ReLU-representable.

We assume first that $r$ is Lipschitz continuous. Following the derivations in Theorem 2 in [11], we can find a sequence of convex piecewise linear functions $r_n$ such that $r_n \to r$ uniformly for $n \to \infty$. Furthermore, by Theorem 2.2 in [6] there exist single layer ReLU networks $\mathcal{NN}_n$ that can represent $r_n$ exactly which proves the claim.

If we relax the Lipschitz assumption, we can again follow the arguments in Theorem 2 in [11] and find a sequence of Lipschitz continuous and convex functions $\hat{r}_k$ that converge uniformly to $r$. Hence for $\epsilon > 0$ we can find $k \geq 1$ such that $\|r - \hat{r}_k\|_\infty < \epsilon/2$. With the arguments from above we can then find $r_n$ such that $\|\hat{r}_k - r_n\|_\infty < \epsilon/2$, and hence $\|r - r_n\|_\infty < \epsilon$ which proves the claim.

### A.8 Restart Procedure for our DCA Routine

Following [46], let the index sets $I_h(x^*)$ and $I_g(x^*)$ be defined as follows

$$I_h(x^*) = \{i \in \{1, \ldots, N_x\} : \alpha_i < 0 \text{ and } \langle w_i, x^* \rangle + b_i = 0\}, \tag{30}$$

$$I_g(x^*) = \{i \in \{1, \ldots, N_x\} : \alpha_i > 0 \text{ and } \langle w_i, x^* \rangle + b_i = 0\}. \tag{31}$$

Note that if $I_h(x^*) = \emptyset$ or $I_g(x^*) = \emptyset$ we have that $\partial h(x^*) \subseteq \partial g(x^*)$ as one of the two sets is a singleton and $x^*$ is DC-critical. Hence, for the rest of this derivation we assume $I_h(x^*) \neq \emptyset$ and $I_g(x^*) \neq \emptyset$.

Now, the subgradients of $h$, respectively $g$, are given by

$$\partial h(x^*) = \left\{ \xi + \sum_{\alpha_i < 0} |\alpha_i| w_i H(\langle w_i, x^* \rangle + b_i) \epsilon_i^h : \epsilon_i^h \in \begin{cases} [0,1] & \text{if } i \in I_h(x^*) \\ \{1\} & \text{else} \end{cases} \right\}, \tag{32}$$

$$\partial g(x^*) = \left\{ \rho x^* + \sum_{\alpha_i > 0} \alpha_i w_i H(\langle w_i, x^* \rangle + b_i) \epsilon_i^g : \epsilon_i^g \in \begin{cases} [0,1] & \text{if } i \in I_g(x^*) \\ \{1\} & \text{else} \end{cases} \right\}, \tag{33}$$

where $H(z)$ denotes the Heaviside function, i.e., $H(z) = 1$ for $z \geq 0$ and $H(z) = 0$ else. Furthermore, let us define the following

$$I_h^+(x^*) = \{i \in \{1, \ldots, N_x\} : \alpha_i < 0 \text{ and } \langle w_i, x^* \rangle + b_i > 0\}, \tag{34}$$

$$I_g^+(x^*) = \{i \in \{1, \ldots, N_x\} : \alpha_i > 0 \text{ and } \langle w_i, x^* \rangle + b_i > 0\}, \tag{35}$$

$$v_h = \xi + \sum_{i \in I_h^+(x^*)} |\alpha_i| w_i, \tag{36}$$

$$v_g = \rho x^* + \sum_{i \in I_g^+(x^*)} \alpha_i w_i, \tag{37}$$

as well as, the matrix $A_h \in \mathbb{R}^{n \times |I_h(x^*)|}$ with columns $|\alpha_i| w_i$ for $i \in I_h(x^*)$, and the matrix $A_g \in \mathbb{R}^{n \times |I_g(x^*)|}$ with columns $\alpha_i w_i$ for $i \in I_g(x^*)$. The inclusion $\partial h(x^*) \subseteq \partial g(x^*)$ holds true if

$$\forall \epsilon^h \in [0,1]^{|I_h(x^*)|} \exists \epsilon^g \in [0,1]^{|I_g(x^*)|} : v_h + A_h \epsilon^h = v_g + A_g \epsilon^g. \tag{38}$$

To check if this holds, we can solve the max-min problem

$$\max_{0 \leq \epsilon^h \leq 1} \min_{0 \leq \epsilon^g \leq 1} \|v_h + A_h \epsilon^h - v_g - A_g \epsilon^g\|_1, \tag{39}$$

and observe whether or not the optimal objective value is zero. If it is, (38) holds and we have found a local minimum. If not, then we have found an element $x_0^* \in \partial h(x^*)$ with $x_0^* \notin \partial g(x^*)$. Following [44], we can restart the DCA routine with $x_0^*$ and yield a strict energy reduction in the first step.

Figure 2: Illustration of Batched DCA for batch size 512

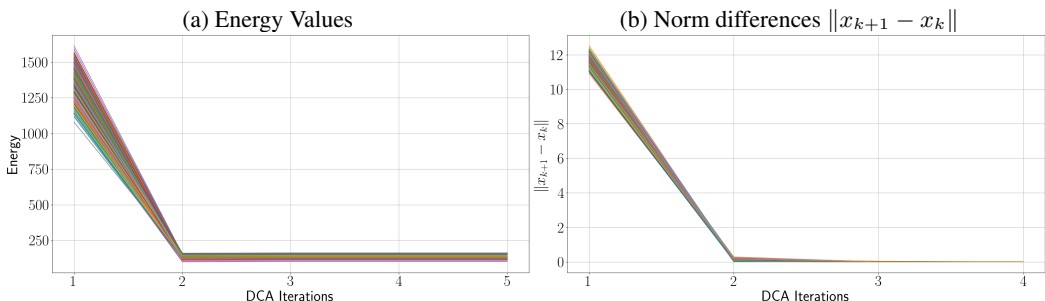

(a) Energy Values  (b) Norm differences $\|x_{k+1} - x_k\|$

*Note.* Illustration of our batched DCA routine computing locally minimal energy states for 512 reasoning tasks in parallel.

What remains to be discussed, is how to solve (39) efficiently. To do so, note that we can reformulate the problem as

$$\max_{\epsilon^h} \min_{\epsilon^g, r_1, r_2} \langle r_1, \mathbb{1} \rangle + \langle r_2, \mathbb{1} \rangle \tag{40}$$

$$\text{s.t.} \ \ v_h + A_h \epsilon^h - v_g - A_g \epsilon^g = r_1 - r_2 \tag{41}$$

$$r_1, r_2, \epsilon^h, \epsilon^g \geq 0 \tag{42}$$

$$\epsilon^h, \epsilon^g \leq 1 \tag{43}$$

This problem can be solved using a branch and bound strategy for bilevel linear programming (see, e.g., section 6.2 in [8]). Furthermore, since $x^*$ is DC-critical, we know that for $\epsilon^h = \mathbb{1}$ there exists an $\epsilon_*^g \in [0,1]^{|I_g(x^*)|}$ with $v_h + A_h \mathbb{1} = v_g + A_g \epsilon_*^g$. Hence, we can warm start the branch and bound routine with $(\epsilon^h, \epsilon^g, r_1, r_2) = (\mathbb{1}, \epsilon_*^g, 0, 0)$. If the optimal objective value is zero, we already start with an optimal solution. If not, we can stop the branch and bound routine as soon as we have found the first $(\epsilon^h, \epsilon^g)$ with $v_h + A_h \epsilon^h \neq v_g + A_g \epsilon^g$. Thus, the restart procedure can be implemented efficiently.

## B  Details on Numerical Experiments

### B.1  Visualization of Convergence

We visualize how our batched DCA routine converges for 512 reasoning tasks in parallel in Figure 2. In particular we used the matrix inversion dataset, see Section 5, and for a fixed $y$ visualized the energy values $E_\theta(x_k, y)$ and norm differences $\|x_{k+1} - x_k\|$ after training the parameters $\theta$, i.e., at inference time.

### B.2  Details on Experimental Setup

For our baselines we closely follow the code releases of [20][7] and [21][8].

**Model Specifications.** For DCAReasoner we set the number of hidden units $N_x = 8$ and $N_y = 4000$ for all benchmark datasets. For our baselines we scale the network size depending on the dataset to ensure that all models have roughly the same number of parameters. We set the convergence tolerance in Algorithm 1 to $10^{-5}$ and set a maximum of 30 DCA iterations. However, in our experiments we see that the batched DCA algorithm consistently converges to machine precision in less than 10 iterations. Hence, we also observe finite convergence guaranteed by Theorem 1 empirically. Starting points $x^0$ are sampled uniformly random as stated in Algorithm 1. We set $l = -1$ and $u = 1$ for all benchmark datasets except for *QR* and *Matrix Multiplication* for which we use $l = -5$ and $u = 5$. Hyperparameters are summarized in Table 3.

---

[7] https://github.com/yilundu/irem_code_release, MIT License
[8] https://github.com/yilundu/ired_code_release, MIT License

| Hyperparameter | Value | | |
|---|---|---|---|
| | DCAReasoner | IREM | IRED |
| Common Hyperparameters | | | |
| Learning Rate | $10^{-4}$ | $10^{-4}$ | $10^{-4}$ |
| Batch Size | 512 | 512 | 512 |
| Number of Gradient Steps | 10.000 | 10.000 | 10.000 |
| Starting Point Sampling | uniform | uniform | normal |
| DCAReasoner Specific Hyperparameters | | | |
| Number of Neurons $N_x$ | 8 | | |
| Number of Neurons $N_y$ | 4000 | | |
| DCA Convergence Tolerance tol | $10^{-5}$ | | |
| Maximum DCA Iterations | 30 | | |

Table 3: Hyperparameters settings for our experiments. Note that for our baselines we scale the network size depending on the dataset to ensure that all models have roughly the same number of parameters.

**Training.** For training, we use the Adam optimizer with a learning rate of $10^{-4}$ as suggested in [20, 21] for all models. We set the batch size to 512 and train each model for a fixed number of 10000 iterations, i.e., gradient steps. Hyperparameters are summarized in Table 3.

**Evaluation.** For evaluation, we are using again a batch size of 512 and perform 20 test iterations, summing up to roughly 10000 test problems per difficulty level, i.e., once in the easy setting and once in the hard setting.

**Hardware.** All experiments are performed on a n1-standard-2 Google cloud instance with 7.5GB RAM and two NVIDIA T4 GPUs.

### B.3 Details on Benchmark Datasets

#### B.3.1 Motivation

In general, neural algorithmic reasoning constitutes an unsolved problem in machine learning. For an argumentation on the complexity of neural algorithmic reasoning see e.g. [47]. Matrix Completion, QR Decomposition, and Matrix Multiplication represent algorithmic reasoning tasks. Du et. al. argue that effective algorithmic reasoning requires repetitive application of underlying algorithmic computations, dependent on problem complexity, and thus serves as a natural benchmark for iterative reasoning [20]. Learning parities is a well-known reasoning benchmark and well-studied in learning theory in general. Shoshani and Shamir argue that there is strong empirical evidence that suggests that parities cannot be learned using more standard general purpose learning methods, and in particular gradient methods, once the dimension is even moderately large [42]. This also goes in line with the fact that our baselines struggled particularly on this task, see also conclusion and limitations section in [20]. For matrix inversion, Ji et. al. argue that despite significant progress in deep learning, there exists no universal neural-based method for approximating matrix inversion [30], showing that this is a non-trivial task for neural reasoning.

#### B.3.2 Technical Details

In the following we give more details on our benchmark datasets. We followed the code releases of [20][7] and [21][8] closely.

**Matrix Inverse.** We construct well-conditioned random $20 \times 20$-matrices $A = 2MM^T + 0.5 \cdot I$, with $M$ being a random matrix with entries in $[-1, 1]$. The task is then to compute the inverse $A^{-1}$. Harder tasks are created by making the matrices less-well conditioned by setting $A = 2MM^T + 0.1 \cdot I$.

**Matrix Completion.** We randomly construct low-rank $20 \times 20$-matrices $A = 0.1 \cdot N + \frac{1}{20}UV^T$ where $N$ is a standard normally distributed noise matrix and $U$ and $V$ are again standard normally distributed random $20 \times 10$-matrices. Then, a random mask is created by rounding a randomly generated uniformly distributed matrix $M$. The model is then presented with the masked matrix $A$ and asked to recover it. Harder tasks are created by setting $A = 0.1 \cdot N + \frac{1}{5}UV^T$.

Figure 3: Empirical Analysis of Robustenss to Noise on the QR Decomposition Dataset

*Note.* We evaluated the trained models on the QR benchmark dataset adding different Gaussian noise levels to the input data. IREM becomes unstable for large noise levels while IRED and DCAReasoner appear to be robust to noisy inputs.

**Parity.** Similar to [25], we create uniformly random vectors in $[0, 1]^{20}$ and then set the target to 0 if the number of values strictly greater than $0.5$ is even and 1 otherwise. Harder tasks are created by drawing random vectors from $[-1, 2]^{20}$.

**QR Decomposition.** We create uniformly random matrices $A$ with entries in $[-1, 1]$ and then compute the QR decomposition, i.e., $A = QR$. The models are then given the matrix $A$ and asked to reconstruct both $Q$ and $R$. Harder tasks are created by creating uniformly random matrices $A$ with entries in $[-2.5, 2.5]$.

**Matrix Multiplication.** We create uniformly random matrices $A$ with entries in $[-1, 1]$ and then compute the square, i.e., $A^2$. The models are then given the matrix $A$ and asked to perform the matrix multiplication $A^2$. Harder tasks are created by creating uniformly random matrices $A$ with entries in $[-1.5, 1.5]$.

### B.4   Further Experiments Analyzing Robustness to Noisy Data

We evaluated the trained models on the QR benchmark dataset adding different Gaussian noise levels to the input matrix before processing with a scale varying in $\{10^{-4}, 10^{-3}, 10^{-2}, 10^{-1}, 1, 10\}$. Results are visualized in Figure 3. It appears that IREM becomes unstable for large noise levels while IRED and DCAReasoner are mostly robust to noise.

### B.5   Details on Energy-Based Reasoning in Token Embedding Space

For our experiments, we make use of the symptom-to-diagnosis dataset for medical reasoning which is freely available on huggingface.[9] It provides a training (853 examples) and test (212 examples) dataset consisting of short texts in which a patient describes her symptoms and a corresponding diagnosis out of a set of 22 medical diagnoses summarized in Table 4.

We first finetune an uncased DistilBERT model for text classification, splitting the training set again into training and validation using a $80/20$ split, and using the huggingface trainer for sequence classification. In particular, we use a batch size of 16, a learning rate of $2 * 10^{-5}$, a weight decay of $0.01$, and 10 evaluation steps saving the best performing model in terms of accuracy (which is also 96% as for the energy-based models).

We then use the CLS token embeddings of the texts in the training dataset of this finetuned model as inputs $y$ and the embeddings of the corresponding diagnosis as a ground truth $x$ to train the DCAReasoner and our baselines in a continuous reasoning setting. The model specifications for DCAReasoner are the same as in our main experiments, i.e., we set $N_x = 8$ and $N_y = 4000$,

---

[9]`https://huggingface.co/datasets/gretelai/symptom_to_diagnosis`, the dataset is licensed under Apache 2.0

| Diagnosis | Training Examples | Test examples |
|---|---|---|
| drug reaction | 40 | 8 |
| allergy | 40 | 10 |
| chicken pox | 40 | 10 |
| diabetes | 40 | 10 |
| psoriasis | 40 | 10 |
| hypertension | 40 | 10 |
| cervical spondylosis | 40 | 10 |
| bronchial asthma | 40 | 10 |
| varicose veins | 40 | 10 |
| malaria | 40 | 10 |
| dengue | 40 | 10 |
| arthritis | 40 | 10 |
| impetigo | 40 | 10 |
| fungal infection | 39 | 9 |
| common cold | 39 | 10 |
| gastroesophageal reflux disease | 39 | 10 |
| urinary tract infection | 39 | 9 |
| typhoid | 38 | 9 |
| pneumonia | 37 | 10 |
| peptic ulcer disease | 37 | 10 |
| jaundice | 33 | 7 |
| migraine | 32 | 10 |

Table 4: Number of examples per split and diagnosis in the Symptom-to-Diagnosis dataset.

Figure 4: Training and Validation Curves for Sudoku Experiment

(a) Training Loss (Cross-Entropy)    (b) Validation Performance (Cell Accuracies)

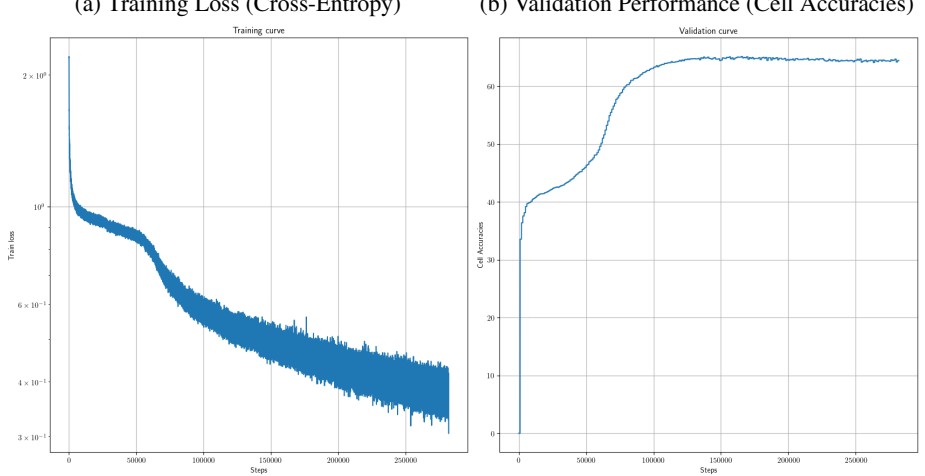

*Note.* Cross-Entropy loss during training (left) and cell accuracies on validation data (right). Validation steps are performed every 1000 training steps.

scaling the network sizes in our baselines to ensure that all models have roughly the same number of parameters.

For training, we use a batch size of $64$ and a learning rate of $10^{-4}$. We train each model for $10000$ iterations. For evaluation, we use the test set and process it again in batches of size $64$.

## C   Preliminary Experiments in Discrete Setting

As mentioned in the main paper, DCAReasoner is designed for continuous reasoning tasks. Thus, our method cannot be applied as-is to experiments in discrete spaces. Nevertheless, we performed

preliminary experiments using the Sudoku dataset in [41]. The authors provide a training dataset consisting of 1.8 million Sudoku puzzles as well as a validation set with 0.1 million Sudoku puzzles.

For reasoning in a discrete setting our energy landscape is defined in the logit space, i.e., a 729 dimensional setting and we replace the MSE with the cross entropy loss. Processing the inputs, i.e., Sudoku puzzles, can be done in multiple ways. We decided for a convolutional neural network with the residual connection design as in [21] to process the Sudoku puzzles before feeding them to our neural network components. Using the same settings as in the main paper, i.e., $N_x = 8$, $N_y = 4000$ and a learning rate of $10^{-4}$, but a batch size of 64, we train DCAReasoner for a total of 10 epochs. Validation steps, i.e., computing the cell accuracy defined as the percentage of unfilled cells whose values are correctly predicted on the validation data [41], are performed every 1000 training steps. The training process is visualized in Figure 4. Note that the validation performance stagnates after roughly 6 epochs. At this time, we observe a cell accuracy of 65.09% , i.e., 65.09% of all unfilled cells in all of the 100K validation puzzles are filled correctly. For context, the baseline from [41] with random order of input cells achieves 52%, while the one with fixed order achieves 58.64%. Our results only fall short from the accuracy achieved by the model receiving additional context information about solving strategies during training (94.23%). However, we point out that the specific architecture of our model does not allow such information to be provided.

