# OpenReview forum: "A Difference-of-Convex Functions Approach to Energy-Based Iterative Reasoning"
_NeurIPS.cc/2025/Conference — NeurIPS 2025 poster_

### Official Review · Reviewer_A3zE · 2025-06-29

**Clarity:** 3
**Significance:** 3
**Originality:** 3
**Rating:** 5
**Confidence:** 4

**Summary:**

This paper proposes an approach to energy-based reasoning where a difference of convex functions is used to represent the energy function. The paper provides theoretical results illustrating how this approach can approximate arbitrarily complex functions, while also ensuring that optimization converges to valid minima. Finally, the paper illustrates the applicability of the method on several numerical tasks, with an interesting energy landscape recovered in token reasoning. Overall, this paper has a nice blend of both emperical practice as well as theoretical analysis

**Questions:**

- Would the method presented in the paper also work with non single layer MLP networks?
- How robust are the energy minima recovered from this approach? Do they have less local
minima than prior work on energy based reasoning?
- Could the authors provide a bit more clarification on why the subgradient in Equation 8 of the paper
is the subgradient of Equation 4? I derived a slightly different form when going through the computation.

**Ethical Concerns:**

["NO or VERY MINOR ethics concerns only"]

**Final Justification:**

I remain positive about this paper and maintain my positive score.

**Limitations:**

The paper adequetly discusses the limitations of the approach.

**Paper Formatting Concerns:**

No concerns

**Quality:**

3

**Strengths And Weaknesses:**

Strengths:

- The paper is clear and well motivated
- The paper provides theoretical justification on their approach, with the approach
having a provable minimum that it can converge to.
- The paper illustrates a set of promising results on real-world settings

Weaknesses

- Most of the setting shown in the paper are a little toy, but I don't think this really
detracts from the paper, as I think it sufficient to illustrate the promise of the method

---

> ### Author Rebuttal · Authors · 2025-07-30
>
> We want to thank the reviewer for the positive feedback on our work. In the following we comment on the points raised by reviewer A3zE.
>
> **"Would the method presented in the paper also work with non single layer MLP networks?"**
>
> Yes, all the networks $\alpha$, $W$, $b$, $\xi$, and $\rho$ can be set to arbitrary differentiable architectures. As such, it would also be possible to train on different modalities of $y$ such as text, images, or graphs and perform continuous reasoning in such settings. See also our preliminary experiments for reviewer P8YW, in which we make use of convolutional neural networks to process Sudoku puzzles as inputs.
>
> **"How robust are the energy minima recovered from this approach? Do they have less local minima than prior work on energy based reasoning?"**
>
> We thank the reviewer for this important question. As both, our baselines and DCAReasoner, are able to represent highly non-convex energy functions it is not possible to give any general statements on the number of local minima. Also projecting the energy landscapes of our trained baselines and DCAReasoner to lower dimensional spaces to compare them empirically can lead to wrong conclusions due to a loss of information in the projection. Nevertheless, we think that the convergence guarantees to local minima in our approach alone makes DCAReasoner more robust than our baselines. We also performed additional experiments for reviewer GYtm showing that our method is robust to noisy inputs and outperforms our baselines even in the presence of large noise levels. Please see our answer to reviewer GYtm for more details on the experiments.
>
> **"Could the authors provide a bit more clarification on why the subgradient in Equation 8 of the paper is the subgradient of Equation 4? I derived a slightly different form when going through the computation."**
>
> The full subgradient of $h$ is given in equation (32) in our manuscript. Equation (8) is merely a specific element of the subgradient that results from setting all $\epsilon_i^h=1$ in equation (32). We give more details on this specific choice later on.
>
> To derive equation (32) we make use of the following properties of the subgradient:
> * For $\lambda \geq 0$, we have $\partial (\lambda f)(x) = \\{\lambda u: u \in \partial f(x)\\}$
> * Sum rule (Proposition 5.4.6 in **[1]**): If $f_1,f_2:\mathbb{R}^n\to (-\infty, \infty]$ are convex functions satisfying $\text{ri}(\text{dom}(f_1)) \cap \text{ri}(\text{dom}(f_2)) \neq \emptyset$, $\text{ri}$ denoting the relative interior, and $f_1+f_2$ is proper, then for all $x\in\mathbb{R}^n$ we have $\partial(f_1+f_2)(x)=\\{u+v: u\in\partial f_1(x), v\in\partial f_2(x)\\}$.
> * Let $g(x) = f(Ax+b)$ with $f:\mathbb{R}\to (-\infty, \infty]$ then $\partial g(x)=A^{\ast}\partial f(Ax+b)$ if the range of $A$ contains a point of the relative interior of $\text{dom}(f)$, see Theorem 23.9 in **[2]**.
>
> Equation (32) then follows directly from the fact that the subgradient of the ReLU activation function is given by $\partial(\text{ReLU})(x)=\\{1\\}$ for $x>0$, $\partial(\text{ReLU})(0)=\[0,1\]$, and $\partial(\text{ReLU})(x)=\\{0\\}$ for $x<0$.
>
> To ensure finite convergence of our DCA routine, Theorem 9 in **[3]** makes use of a so called fixed choice of subgradients which is defined in section 2.4.2 in **[3]**. This fixed choice is formalized via a mapping from the subgradient of $h$ in a specific point (set-valued) to a specific element therein. In our case, this mapping is given by always choosing the element in which all $\epsilon_i^h=1$ resulting in equation (8), thus avoiding to check exactly which of the affine terms evaluate to zero (and would allow for $\epsilon_i^h\in[0,1]$) and having a rather simple form using the properties of the heaviside function, i.e., $H(z)=1$ for $z\geq 0$ and $H(z)=0$ else.
>
> **References**
>
> **[1]** Bertsekas, Dimitri. Convex optimization theory. Vol. 1. Athena Scientific, 2009.
>
> **[2]** Rockafellar, Ralph T. Convex Analysis. Princeton Mathematical Series Princeton University Press, Princeton, N. J., (1970)
>
> **[3]** Thi Hoai An, Le, and Pham Dinh Tao. "Solving a class of linearly constrained indefinite quadratic problems by DC algorithms." Journal of global optimization 11.3 (1997): 253-285

---

### Official Review · Reviewer_P8YW · 2025-07-02

**Clarity:** 3
**Significance:** 2
**Originality:** 3
**Rating:** 4
**Confidence:** 3

**Summary:**

The paper presents a novel approach for iterative reasoning that formulates the problem of reasoning as energy minimization, which, with appropriate neural network parameterization, is transformed into a difference of convex functions. The authors provide theoretical convergence guarantees as well as expressivity results for the parameterization used. Finally, the DCA routine is scaled to perform batched optimization. The experiments are performed on matrix-based tasks (completion, inversion, decomposition, etc) and energy landscape optimization in the embedding space of language models to show the efficacy of the proposed method compared to previous approaches for energy-based iterative reasoning.

**Questions:**

1. What is the motivation behind selecting the task in section 5.3? The paper can be made significantly more impactful if a more realistic task, ideally in discrete space (for example, Sudoku or Zebra Puzzles), is chosen to demonstrate the effectiveness of the proposed method. In that setting, it would be nice to compare the performance of DCAReasoner with iterative feed-forward approaches like discrete diffusion and search-based auto-regressive methods (for example, [1,2])

2. As I understand from the description, the DistilBERT was fine-tuned on the symptom-to-diagnosis dataset (please correct me if I'm misinterpreting here). If so, what was the accuracy of the DistilBERT classifier?

# References
[1] [[2409.10502] Causal Language Modeling Can Elicit Search and Reasoning Capabilities on Logic Puzzles](https://arxiv.org/abs/2409.10502)

[2] [[2502.06768] Train for the Worst, Plan for the Best: Understanding Token Ordering in Masked Diffusions](https://arxiv.org/abs/2502.06768)

**Ethical Concerns:**

["NO or VERY MINOR ethics concerns only"]

**Final Justification:**

Given how toyish the other tasks in the experiments are, my concern regarding the paper's significance still remains. However, given the paper's other strengths and the preliminary results on the Sudoku task, I'm updating my score.

**Limitations:**

yes

**Quality:**

3

**Strengths And Weaknesses:**

**Quality**: The authors provide theoretical convergence guarantees for DCAReasoner as well as expressivity results for the parameterization used. The experiments demonstrate the improvement in efficiency compared to previous energy-based reasoning approaches. However, the choice of tasks used in the experiments and the lack of comparison with other kinds of baselines (see question 1 below) fall short in providing a comprehensive picture.

**Significance**: While the general idea of energy-based iterative reasoning is quite promising. The choice of experiments and baselines could be expanded to make the paper more impactful.

**Clarity**: The paper is well written and easy to follow.

**Originality**: The use of DC optimization for energy-based iterative reasoning and its proof of convergence both incorporate original ideas.

---

> ### Author Rebuttal · Authors · 2025-07-31
>
> In the following we comment on the points raised by reviewer P8YW.
>
> **"What is the motivation behind selecting the task in section 5.3? The paper can be made significantly more impactful if a more realistic task, ideally in discrete space (for example, Sudoku or Zebra Puzzles), is chosen to demonstrate the effectiveness of the proposed method. In that setting, it would be nice to compare the performance of DCAReasoner with iterative feed-forward approaches like discrete diffusion and search-based auto-regressive methods (for example, [1,2])"**
>
> Section 5.3 shows that our method is able to learn reasonable energy landscapes in token embedding space. That is, our energy landscape is able to capture fine-grained textual information when added on top of a transfomer based architecture. We think that DCAReasoner might be used to improve reasoning skills of LLMs by adding it as a corrective component (in a continuous sense) between the last hidden state and the LLM head. Of course the details of such an energy guided text generation approach are out of scope for our current work. Thus, section 5.3 should merely motivate such an approach and show that our baselines would not be suitable due to much higher inference times.
>
> As mentioned in our manuscript, DCAReasoner is designed for continuous reasoning tasks. Thus, our method cannot be applied as-is to experiments in discrete spaces. This is also why the task in section 5.3 was formulated as a continuous reasoning task instead of a classification task. Nevertheless, we performed preliminary experiments using the Sudoku dataset in **[1]** suggested by the reviewer. For reasoning in a discrete setting our energy landscape is defined in the logit space, i.e., a 729 dimensional setting and we replace the MSE with the cross entropy loss. Processing the inputs, i.e., sudoku puzzles, can be done in multiple ways. We decided for a convolutional neural network with the residual connection design as in **[2]** to process the Sudoku puzzles before feeding them to our neural network components. Using the same settings as in the main paper ($N_x=8$, $N_y=4000$, a learning rate of $10^{-4}$) but a batch size of 64, we observe a cell accuracy of 65.09% after training for 6 epochs, i.e., 65.09% of all unfilled cells in all of the 100K test puzzles are filled correctly. For context, the baseline from **[1]** with random order of input cells achieves 52%, while the one with fixed order achieves 58.64%. Our results only fall short from the accuracy achieved by the model receiving additional context information about solving strategies during training (94.23%). However, we point out that the specific architecture of our model does not allow such information to be provided.
>
> We will include a section into the appendix on how DCAReasoner might be used in a discrete setting for future research and report our early results. Nevertheless, we want to point out again that DCAReasoner as presented in our manuscript is designed for continuous reasoning tasks.
>
> **"As I understand from the description, the DistilBERT was fine-tuned on the symptom-to-diagnosis dataset (please correct me if I'm misinterpreting here). If so, what was the accuracy of the DistilBERT classifier?"**
>
> Yes, this is correct. This step should ensure that the CLS token embeddings that are used within our energy-based reasoning approaches are encoding the necessary information for the classification task in this dataset. The accuracy of the DistilBERT classifier is mentioned in Appendix B.4 and is 96% as for the energy-based baselines. We note again that the aim of this experiment was not to train a better classifier but to show that DCAReasoner is able to learn reasonable energy landscapes in token embedding space while improving inference time. In a potential camera-ready version we will include an additional clarifying comment on the motivation for this experiment.
>
> **References**
>
> **[1]** Shah, K., Dikkala, N., Wang, X., & Panigrahy, R. (2024). Causal language modeling can elicit search and reasoning capabilities on logic puzzles. Advances in Neural Information Processing Systems (NeurIPS), 37, 56674-56702.
>
> **[2]** Y. Du, J. Mao, and J. Tenenbaum. Learning iterative reasoning through energy diffusion. International Conference on Machine Learning (ICML), pages 11764–11776, 2024.

---

### Official Review · Reviewer_GYtm · 2025-07-05

**Clarity:** 3
**Significance:** 4
**Originality:** 4
**Rating:** 5
**Confidence:** 4

**Summary:**

This paper introduces DCAReasoner, an energy-based iterative reasoning algorithm leveraging a Difference-of-Convex decomposition. It provides finite-step convergence guarantees and universal function approximation, and achieves 3–27× inference speedups on tasks including polynomial root finding, matrix inversion, and quadratic programming, while matching or exceeding state-of-the-art accuracy.

**Questions:**

* The acronym “DCA” appears before it is explicitly defined. Could you please clarify and provide its full form at first use?
* What criteria guided your choice of the number of DCA iterations K in each experiment? How sensitive are your results—both in terms of accuracy and convergence speed—to different values of K and to the initialization x_0?
* The paper does not fully specify the hyperparameters and configurations used for the baseline methods. Could you include detailed settings for all baselines to ensure reproducibility and a fair comparison?

**Ethical Concerns:**

["NO or VERY MINOR ethics concerns only"]

**Final Justification:**

The authors had addressed my concerns, so I keep my positive rating.

**Limitations:**

Yes

**Quality:**

4

**Strengths And Weaknesses:**

## Strengths

* Theoretical guarantees. Finite-step convergence to a local minimum and universal approximation results under mild sign-constraints ensure both reliability and expressivity.
* End-to-end trainability. By deriving closed-form updates for each DC subproblem, the entire K-step inference can be unrolled into a differentiable computation graph, enabling direct gradient-based optimization of the energy model parameters.
* Fast training. Closed-form DC updates replace inner gradient-descent loops, reducing both per-epoch compute and memory overhead—training runs substantially faster than methods that nest gradient-descent in the forward pass.
* Significant inference speedups. Achieves 3–27× faster inference on polynomial root finding, matrix inversion, and QP benchmarks while matching or exceeding prior accuracy.
* Broad applicability. Demonstrated across continuous algorithmic reasoning, high-dimensional embedding-space classification, and convex/non-convex energy landscapes.

## Weakness
* Generalization Beyond Training Distribution: Because the energy function $E_\theta(x;y) $ is learned to minimize MSE on the training pairs $(y_i,x_i)$, it may overfit to those specific problem instances. It’s unclear how well it generalizes to out‐of‐distribution y (e.g.\ different matrix condition numbers or QP formulations).
* High-Dimensional Scalability: Solving the DC subproblem still requires inverting a Hessian-like operator or solving a fairly large convex system at each step. While tractable up to a few hundred dimensions, it may become prohibitive in, say, $\geq10^4$-dimensional settings without further approximations.
* Robustness to Noise and Model Mis-specification: There is no evaluation of how DCAReasoner handles noisy or approximate y (e.g.\ measurement error in the input matrix) or errors in the DC decomposition. Such perturbations could cause the DCA iterates to converge to spurious local minima.

---

> ### Author Rebuttal · Authors · 2025-07-30
>
> We want to thank the reviewer for the positive feedback on our work. In the following we comment on the points raised by reviewer GYtm.
>
> **"Generalization Beyond Training Distribution: Because the energy function $E_\theta(x;y)$ is learned to minimize MSE on the training pairs $(y_i, x_i)$, it may overfit to those specific problem instances. It’s unclear how well it generalizes to out‐of‐distribution $y$ (e.g.\ different matrix condition numbers or QP formulations)."**
>
> To test the generalization on out-of-distribution samples our evaluation in Table 1 includes not only the performance on the test set of the same distribution but also on a task-specific harder test set following **[1]** and **[2]**. Details on how we increase the difficulty for each task in this test set are reported in Appendix B.3. For instance, the matrix inverse task follows exactly the reviewer's suggestion of increasing the difficulty by making the input matrix less well-conditioned.
>
> **"High-Dimensional Scalability: Solving the DC subproblem still requires inverting a Hessian-like operator or solving a fairly large convex system at each step. While tractable up to a few hundred dimensions, it may become prohibitive in, say, $\ge 10^4$-dimensional settings without further approximations."**
>
> While it is true that the solution of the convex system requires inverting a matrix, Lemma 4 shows that this matrix is merely a scaled identity, i.e., $\rho I$, where $\rho$ is a positive scalar. Thus, the inverse can be computed analytically. That is, in line 7 of Algorithm 1 the solution of the mentioned convex system is already encoded by the factor $1/\rho$ which allows our algorithm to scale well in high-dimensional settings. We will include further explanations on Algorithm 1 in section 4.2 in a potential camera-ready version.
>
> **"Robustness to Noise and Model Mis-specification: There is no evaluation of how DCAReasoner handles noisy or approximate y (e.g.\ measurement error in the input matrix) or errors in the DC decomposition. Such perturbations could cause the DCA iterates to converge to spurious local minima."**
>
> We thank the reviewer for raising this important point. We evaluated the trained models on the QR benchmark dataset adding different Gaussian noise levels to the input matrix before processing with a scale varying in $\\{10^{-4},10^{-3},10^{-2},10^{-1},1,10\\}$. Due to restrictions in the NeurIPS review process we can merely present the outcome of our experiments in a markdown table, but we would include proper visualizations in the appendix of a potential camera-ready version. The table below shows the MSEs for different noise levels. It appears that IREM becomes unstable for larger noise levels, while IRED and DCAReasoner are mostly robust to noise.
>
> |             | $10^{-4}$   | $10^{-3}$   | $10^{-2}$   | $10^{-1}$   | $1$    | $10$   |
> | ----------- | ----------- | ----------- | ----------- | ----------- | ------ | ------ |
> | DCAReasoner | 0.1439      | 0.1436      | 0.1439      | 0.1453      | 0.1980 | 0.7833 |
> | IRED        | 0.2176      | 0.2173      | 0.2174      | 0.2181      | 0.2646 | 0.9196 |
> | IREM        | 0.152       | 0.1521      | 0.1522      | 0.1533      | 0.444  | 15.9426|
>
> **"The acronym “DCA” appears before it is explicitly defined. Could you please clarify and provide its full form at first use?"**
>
> We thank the reviewer for pointing this out. We will clarify and provide the full meaning of DCA, Difference-of-Convex function algorithm **[3]**, at it's first appearance in a potential camera ready version.
>
> **"What criteria guided your choice of the number of DCA iterations K in each experiment? How sensitive are your results—both in terms of accuracy and convergence speed—to different values of K and to the initialization x_0?"**
>
> In all our experiments the DCA subroutine converged to machine precision in less than or around 10 iterations. Thus, setting the maximum number of DCA iterations to 30 is merely a safety net if for example numerical issues lead to convergence issues in one of the gradient steps and hence would end up in an infinite loop if there is no strict upper bound. Nevertheless, we did not experience such issues in any of our experiments. Thus, our results are independent of the specific setting of the maximal number of iterations. Of course if one would choose an extremely low number, e.g. 2, it would not allow the DCA routine to converge to something meaningful but this does not represent an actual limitation in our opinion. Initializations of $x_0$ are done uniformly random at training, as well as test time, as our DCA routine converges independently of the actual starting point. There might be an issue, though,  if the targets are very far from the bounds of the uniform initialization, e.g., if the target is 10 and one initializes randomly between -1 and 1. In such a case the DCA routine might need many iterations to reach the desired target. However, in a well-posed machine learning problem with scaled inputs and outputs such a situation can be ruled out.
>
> **"The paper does not fully specify the hyperparameters and configurations used for the baseline methods. Could you include detailed settings for all baselines to ensure reproducibility and a fair comparison?"**
>
> All hyperparameters including the learning rate, the batch size, the number of gradient steps, and lower and upper bounds for random initialization are given in Appendix B.2. The size of the networks for our baselines is scaled to make sure that each of the networks has roughly the same number of parameters as the DCAReasoner with a size determined by $N_x=8$ and $N_y=4000$. The architecture follows exactly those in **[1]** and **[2]**. However, we agree with the reviewer that having all hyperparameters only in text might not give enough visibility. Hence, we will further include a table with all the necessary information in a potential camera-ready submission.
>
> **References**
>
> **[1]** Y. Du, S. Li, J. Tenenbaum, and I. Mordatch. Learning iterative reasoning through energy minimization. International Conference on Machine Learning (ICML), pages 5570–5582, 2022
>
> **[2]** Y. Du, J. Mao, and J. Tenenbaum. Learning iterative reasoning through energy diffusion. International Conference on Machine Learning (ICML), pages 11764–11776, 2024.
>
> **[3]** L. T. H. An and P. D. Tao. The DC (difference of convex functions) programming and DCA revisited with DC models of real world nonconvex optimization problems. Annals of Operations Research, 133:23–46, 2005.

---

> > ### Comment · Reviewer_GYtm · 2025-08-07
> >
> > Thanks for the response. I would like to keep my rating.

---

### Official Review · Reviewer_zLaS · 2025-07-05

**Clarity:** 3
**Significance:** 2
**Originality:** 3
**Rating:** 5
**Confidence:** 3

**Summary:**

The authors proposed a new algorithm named DCAReasoner for energy-based continuous iterative reasoning. It is build upon a tailored class of energy functions for which derived theoretical approximation guarantees. In addition, the authors presented theoretical convergence guarantees for the inherent DCA routine. That is, the authors showed that it converges to local minima in finitely many steps independent of the starting point. Empirically, the authors have proven that it yields improved performance and inference times.

**Questions:**

see Weaknesses

**Ethical Concerns:**

["NO or VERY MINOR ethics concerns only"]

**Final Justification:**

The authors' rebuttal addressed my concerns!

**Limitations:**

YES

**Quality:**

3

**Strengths And Weaknesses:**

Strengths

1. The authors introduce a tailored form of energy functions and present a DCA routine for powering novel energy learning algorithm.

2. The authors derive theoretical convergence guarantees of our DCA routine to local solutions.

3. The authors show that our DCA routine converges in finitely many steps and, hence, offers a clear termination criterion.

4. Under additional assumptions, the authors show how the energy learning algorithm can be scaled for batch optimization and present theoretical approximation guarantees for our form of energy function.

Weaknesses

1. Restrictive Assumption Limiting Expressiveness
The paper's core practical contribution relies on Assumption 1 (requiring all αᵢ ≤ 0), which significantly constrains the model's expressiveness. While the authors provide theoretical approximation guarantees for "convexly-regular" functions, this is a somewhat artificial class they define. The restriction to non-positive weights in the energy function may limit the model's ability to learn complex reasoning patterns that require both positive and negative activations. The authors acknowledge this could affect universal approximation properties but don't thoroughly investigate the practical implications of this limitation.

2. Limited Experimental Scope and Generalization. The experimental evaluation is quite narrow:

	•	Small-scale problems: All continuous reasoning tasks involve only 20×20 matrices, which doesn't demonstrate scalability to realistic problem sizes

	•	 Artificial benchmarks: The tasks (matrix inversion, completion, etc.) are relatively simple mathematical operations that may not reflect the complexity of real-world reasoning scenarios

	•	Token embedding experiment is preliminary: The medical diagnosis task, while interesting, is quite simple (22 classes) and doesn't convincingly demonstrate the method's potential for complex language reasoning applications

---

> ### Author Rebuttal · Authors · 2025-07-30
>
> In the following we comment on the points raised by reviewer zLaS.
>
> **"Restrictive Assumption Limiting Expressiveness: The authors acknowledge this could affect universal approximation properties but don't thoroughly investigate the practical implications of this limitation."**
>
> We would like to point out that our sign constraints are only applied to the output layer, thus making $h$ an input convex neural network (ICNN) in $x$ which are widely used to approximate convex functions in practice. In addition, our experiments show that we outperform or are on par with state-of-the-art energy based reasoning algorithms on multiple benchmark datasets. Given that those baselines offer universal approximation guarantees in theory, we think our results -- and the wide application of ICNNs to approximate convex functions in general -- rule out concerns of practical implications.
>
> **"Small-scale problems: All continuous reasoning tasks involve only 20×20 matrices, which doesn't demonstrate scalability to realistic problem sizes"**
>
> We want to point out that the dimensions were chosen following **[1]** and **[2]** and do not constitute an upper bound. That said, our energy learning framework can potentially scale to even larger settings. Nevertheless, in our opinion reasoning in the space of $20\times20$ matrices which constitutes a 400 dimensional reasoning space is non-trivial. Furthermore, our experiments in token embedding space which are 768 dimensional and the QR decomposition benchmark dataset which is 800 dimensional (two $20\times20$ matrices as targets) show that our approach scales well.
>
> **"Artificial benchmarks: The tasks (matrix inversion, completion, etc.) are relatively simple mathematical operations that may not reflect the complexity of real-world reasoning scenarios"**
>
> We acknowledge that the motivation behind the choice of our benchmark datasets was maybe not elaborated enough in our manuscript so far.
>
> We specifically chose our benchmarks to (i) show how our energy learning framework performs on tasks on which IREM from **[1]** and IRED from **[2]** struggled most, and (ii) to test different aspects of reasoning on well-established benchmark tasks.
>
> In general, neural algorithmic reasoning constitutes an unsolved problem in machine learning. For an argumentation on the complexity of neural algorithmic reasoning see e.g. **[3]**. Matrix Completion, QR Decomposition, and Matrix Multiplication represent algorithmic reasoning tasks. Du et. al. argue that effective algorithmic reasoning requires repetitive application of underlying algorithmic computations, dependent on problem complexity, and thus serves as  a natural benchmark for iterative reasoning **[1]**. Learning parities is a well-known reasoning benchmark and well-studied in learning theory in general. Itamar and Shamir argue that there is strong empirical evidence that suggests that parities cannot be learned using more standard general purpose learning methods, and in particular gradient methods, once the dimension is even moderately large **[4]**. This also goes in line with the fact that our baselines struggled particularly on this task, see conclusion and limitations section in **[1]**. For matrix inversion, Yuliang et. al. argue that despite significant progress in deep learning, there exists no universal neural-based method for approximating matrix inversion **[5]**, showing that this is by no means a trivial task for neural reasoning.
>
> We will add a more elaborate justification for our benchmarks to our Appendix B.3 in a potential camera-ready version.
>
> **"Token embedding experiment is preliminary: The medical diagnosis task, while interesting, is quite simple (22 classes) and doesn't convincingly demonstrate the method's potential for complex language reasoning applications"**
>
> We agree with the reviewer that those experiments should not be seen as a fully developed energy-guided language reasoning application, which is also stated in Section 5.3 as this is not the scope of this work. Our experiments were merely chosen to demonstrate that DCAReasoner is able to learn meaningful energy landscapes in token embedding space, which we believe the outcome suggests.
>
> **References**
>
> **[1]** Y. Du, S. Li, J. Tenenbaum, and I. Mordatch. Learning iterative reasoning through energy minimization. International Conference on Machine Learning (ICML), pages 5570–5582, 2022
>
> **[2]** Y. Du, J. Mao, and J. Tenenbaum. Learning iterative reasoning through energy diffusion. International Conference on Machine Learning (ICML), pages 11764–11776, 2024.
>
> **[3]** P. Veličković, C. Blundell. Neural algorithmic reasoning, Patterns, Volume 2, Issue 7, 100273, 2021
>
> **[4]** S., Itamar, and O. Shamir. Hardness of learning fixed parities with neural networks. arXiv preprint arXiv:2501.00817 (2025).
>
> **[5]** J., Yuliang, J. Wu, and Y. Xi. Rethinking Neural-based Matrix Inversion: Why can't, and Where can. The 28th International Conference on Artificial Intelligence and Statistics (AISTATS), 2025

---

> > ### Comment · Reviewer_zLaS · 2025-08-07
> >
> > Thanks for the response, which addressed my concerns. I will raise my score to 5.

---

> > > ### Author Response · Authors · 2025-08-08
> > >
> > > Thank you for your response. We are happy that we were able to address your concerns.

---

### Decision · Program_Chairs · 2025-09-17

**Decision:**

Accept (poster)

**Comment:**

This paper introduces DCAReasoner, an energy-based iterative reasoning algorithm based on difference-of-convex functions. It introduces a tailored form of energy functions, derived theoretical convergence guarantees, and show that the DCA routine converges in finitely many steps. For larger-sized problems, it provides theoretical approximation guarantees under additional assumptions. Experiments on 5 datasets on the prior benchmark show that the method achieves competitive performance with 1 order-of-magnitude smaller inference time.

The reviewers universally recognize that the method is novel, provides theoretical guarantees, has significant inference-time speedups, and is well-written, which I concur. On the flip side, reviewers raised concerns about the additional assumptions and that the experiments are relatively small-scale. Reviewer GYtm also raised questions about the method's performance under noise and model mis-specification During the rebuttal, the authors added experiments demonstrating that the proposed method is robust to noise compared to baseline methods. The authors also performed preliminary experiments using the Sudoku dataset. Overall the rebuttal addresses most of the reviewers' concerns. During the camera-ready, the authors are encouraged to incorporate the added experiments and discussion into the paper. Also, additional, larger-scale experiments are recommended to demonstrate the scalability of the method.